# Misregulation of the Ubiquitin–Proteasome System and Autophagy in Muscular Dystrophies Associated with the Dystrophin–Glycoprotein Complex

**DOI:** 10.3390/cells14100721

**Published:** 2025-05-15

**Authors:** Manuela Bozzi, Francesca Sciandra, Maria Giulia Bigotti, Andrea Brancaccio

**Affiliations:** 1Dipartimento di Scienze Biotecnologiche di Base, Cliniche Intensivologiche e Perioperatorie, Sezione di Biochimica e Biochimica Clinica, Università Cattolica del Sacro Cuore, Largo F. Vito 1, 00168 Roma, Italy; 2Istituto di Scienze e Tecnologie Chimiche “Giulio Natta”—SCITEC (CNR), Largo F. Vito, 00168 Roma, Italy; francesca.sciandra@cnr.it; 3Bristol Heart Institute, Bristol Royal Infirmary, Research Floor Level 7, Bristol BS2 8HW, UK; g.bigotti@bristol.ac.uk; 4School of Biochemistry, University of Bristol, Bristol BS8 1TD, UK

**Keywords:** dystrophin, dystroglycan, dystroglycanopathies, proteasome degradation, autophagy

## Abstract

The stability of the sarcolemma is severely impaired in a series of genetic neuromuscular diseases defined as muscular dystrophies. These are characterized by the centralization of skeletal muscle syncytial nuclei, the replacement of muscle fibers with fibrotic tissue, the release of inflammatory cytokines, and the disruption of muscle protein homeostasis, ultimately leading to necrosis and loss of muscle functionality. A specific subgroup of muscular dystrophies is associated with genetic defects in components of the dystrophin–glycoprotein complex (DGC), which plays a crucial role in linking the cytosol to the skeletal muscle basement membrane. In these cases, dystrophin-associated proteins fail to correctly localize to the sarcolemma, resulting in dystrophy characterized by an uncontrolled increase in protein degradation, which can ultimately lead to cell death. In this review, we explore the role of intracellular degradative pathways—primarily the ubiquitin–proteasome and autophagy–lysosome systems—in the progression of DGC-linked muscular dystrophies. The DGC acts as a hub for numerous signaling pathways that regulate various cellular functions, including protein homeostasis. We examine whether the loss of structural stability within the DGC affects key signaling pathways that modulate protein recycling, with a particular emphasis on autophagy.

## 1. Introduction

Muscular dystrophies are genetic degenerative diseases that progressively lead to muscle weakness and atrophy. From a clinical and pathological perspective, muscular dystrophies are characterized by different ages of onset and a wide spectrum of phenotypes, including the most severe forms that also involve other organs and tissues, especially the central nervous system.

A subgroup of muscular dystrophies linked to genetic defects hitting the members of the so-called dystrophin–glycoprotein complex (DGC) has been identified. The central component of the DGC is dystrophin, an extended cytoskeletal protein that plays an essential role in the stability of the whole protein complex and that is associated with a series of transmembrane and cytoskeletal proteins [1,2]. Dystrophin is composed of four domains: the N-terminal domain, a very large central domain containing 24 spectrin-like repeats; a cysteine-rich domain, and the C-terminal domain. The protein is strictly connected to the actin cytoskeleton through its N-terminal domain, but further binding epitopes within the central domain reinforce this interaction [3]. Located at its C-terminus, the dystrophin cysteine-rich domain establishes an interaction with the transmembrane protein β-dystroglycan, which in turn non-covalently binds to the extracellular α-dystroglycan [4,5]. Dystroglycan is encoded by the *DAG1* gene as a single polypeptide chain that is post-translationally cleaved into the two dystroglycan subunits α and β, forming a heterodimeric complex (Figure 1) [6].

In skeletal and cardiac muscle, along with integrins, the DGC represents a major transmembrane hub, providing an interaction site for several extracellular matrix binding partners such as laminins, perlecan, agrin, and others. Laminin is one of the most abundant proteins within the specialized basement membrane surrounding muscular fibers and forms a cross-shaped trimer that is composed by the α2, β1, and γ1 subunits (called laminin-211) in muscle. Laminin-211 is linked to α-dystroglycan through an interaction that strongly depends on the presence of calcium divalent cations. This interaction is established between the C-terminal laminin G-like (LG) domains present in laminin or other binding partners and the complex carbohydrate polymer (known as *matriglycan*) protruding from the central mucin-like domain of α-dystroglycan. Thus, the molecular bridge, formed by laminin-211, dystroglycans, and dystrophin, assures a direct connection between the extracellular matrix and the actin cytoskeleton [7].

In muscle, other key members of the DGC are represented by the sarcoglycans, single-pass transmembrane proteins, which are gathered in an extracellular co-folding assembly. This tower-like structure, composed by β-, γ-, and δ-sarcoglycans, provides binding sites for α-sarcoglycan, dystroglycan, and sarcospan (which lies close to β-dystroglycan), as observed in two recent cryo-electron microscopy structural studies [8,9] (Figure 1).

The DGC supplies structural support to the plasma membrane, preventing possible damage generated by contractile muscle activity. At the same time, it serves a key function in the transduction of extracellular stimuli, including mechanical forces, which the DGC converts into biochemical signals [10,11]. The importance of the roles played by the whole membrane-associated DGC is underlined by the fact that many neuromuscular diseases are associated with genetic defects in the crucial constituents of the DGC, such as dystrophin, dystroglycan, laminin, and sarcoglycans. Among them, there are Duchenne muscular dystrophy (DMD), which depends on defects of the dystrophin gene, and congenital muscular dystrophy (CMD), which is linked to mutations hitting laminin α2. Other forms are severe, or later-onset limb–girdle muscular dystrophies (LGMDs), which could also arise from mutations in the dystroglycan (dystroglycanopathies) and in the sarcoglycan (sarcoglycanopathies) genes. Severe Muscle–Eye–Brain (MEB) disease and Walker–Warburg syndrome can also originate from dystroglycan mutations (primary dystroglycanopathies) [12], while a plethora of congenital or milder conditions, collectively called secondary dystroglycanopathies, can arise from mutations hitting several accessory enzymes, located in the endoplasmic reticulum (ER) and Golgi, which are crucial for α-dystroglycan glycosylation.

A common characteristic of these pathologies is the impaired stability of the sarcolemma and its diminished resilience to mechanical stress. As a consequence, uncontrolled fluxes of intracellular calcium ions often trigger cascades of cytosolic reactions, resulting in myofiber necrosis and chronic inflammation. The replacement of myofibers by connective and adipose tissue is often observed in patients that experience progressive weakness and the degeneration of skeletal muscles [13]. In healthy cells, protein synthesis and degradation are finely tuned to maintain protein homeostasis. Proteins are subjected to continuous surveillance, and damaged or unnecessary proteins are removed by the two principal protein degradation systems, i.e., the ubiquitin–proteasome system (UPS) based on the activity of the proteasome 26S and autophagy based on the degradative activity of lysosome. In muscular dystrophies, protein synthesis and degradation are severely unbalanced, and this review focuses on the biochemical processes that lead to the loss of protein homeostasis and muscle mass in each of the main muscular dystrophies related to the DGC.

## 2. Duchenne Muscular Dystrophy

Duchenne Muscular Dystrophy (DMD) occurs due to defects in the gene encoding for dystrophin and is the most common muscular dystrophy in children, with an incidence of about 1 in 5000 live males every year [14]. It features a progressive skeletal muscle weakness, leading to death by the third decade of life, usually following respiratory and/or cardiac failure [15]. Becker Muscular Dystrophy (BMD) is a milder variant of muscular dystrophy caused by genetic defects of the dystrophin gene and characterized by late onset and slower progression [16]. The difference between DMD and BMD lies in the specific mutations affecting the dystrophin gene. DMD is primarily caused by out-of-frame mutations, while BMD results from in-frame mutations [17].

Defective dystrophin, its truncation, or its complete loss destabilize the whole DGC, causing a strong reduction in its constituents, including dystroglycan [18]. Consequently, sarcolemma becomes extremely fragile and not capable of sustaining the stress produced by continuous contraction and relaxation cycles. In this pathological condition, various cellular functions are severely compromised, contributing to disease progression in a synergistic way. For instance, it is well known that protein synthesis and protein degradation, two mutually regulated processes, are unbalanced, and this strongly contributes to muscle atrophy [19]. In particular, the lack of dystrophin causes the DGC to partly or entirely disassemble and/or misfold and directs its components toward intracellular breakdown. For this reason, many therapeutic approaches aim to restore protein homeostasis by rescuing the correct equilibrium between protein synthesis and degradation.

### 2.1. The Upregulation of the UPS and Its Therapeutic Modulation/Inhibition

The proteasome is a multi-subunit protein complex found in the cytosol that plays a crucial role in degrading damaged proteins through its proteolytic activity (Figure 2). It consists of a barrel-shaped core made up of four stacked rings. The two inner rings house the proteolytic subunits responsible for breaking down proteins, while the outer rings interact with regulatory particles. These regulatory particles recognize proteins marked with polyubiquitin chains that indicate they are destined for degradation (Figure 2).

Many investigations focused on the increased activity of the proteasome and calpains (calcium-activated non-lysosomal neutral proteases) in *mdx* mice, a naturally occurring murine animal model for DMD, and in human patients [20,21,22,23]. Indeed, the production of misfolded and/or truncated mutant variants of dystrophin strongly triggers the protein quality control system that protects cells from any potential damage caused by defective proteins. These findings stimulated further studies demonstrating the beneficial effects of molecules that inhibit proteasome or calpain activity, with the rationale behind this approach being the possibility of rescuing, at least partly, dystrophin or DGC components from full degradation and total loss of functionality. Indeed, drugs such as Bortezomib and MG-132 restored the expression levels of DGC members and repressed the pro-inflammatory response, ameliorating the histopathological features of dystrophic skeletal muscles, especially in animal models of DMD [24,25,26,27,28,29,30].

However, a prolonged rescue of the normal phenotype driven by proteasome inhibitors cannot be always observed in tissues from DMD patients [31]. Not surprisingly, in the long term, treatment with these drugs has proven to be ineffective, if not detrimental, since a prolonged proteasome inhibition alters protein homeostasis with deleterious consequences such as the accumulation of misfolded and damaged proteins that can lead to apoptosis and necrosis [32,33,34].

Undesirable side effects, including painful neuropathy, thrombocytopenia, hypotension, and fatigue, have also been reported in patients affected by relapsing multiple myeloma and mantle cell lymphoma, two diseases for which Bortezomib is currently used [35,36,37]. The possible use of proteasome inhibitors seems to be even more contradictory in the study carried out by Wadosky and colleagues on a canine model of muscular dystrophy. These authors found that in this model, proteasome activity was increased in only half of the muscle analyzed, while, for instance, in the heart, many proteins involved in proteolytic processing were found to be downregulated [38]. Such a contradictory biochemical scenario calls into question the basic rationale of any therapeutic approach based on proteasome inhibitors, which—to the best of our knowledge—have not yet been approved for clinical therapy in patients affected by muscular dystrophy.

#### 2.1.1. Inhibition of Muscle Specific Ubiquitin-Conjugating Enzymes

Ubiquitination, i.e., the labeling of misfolded and inactive proteins with ubiquitin (a small protein of about 9 kDa which becomes covalently linked to its target protein) is a fundamental step enabling further downstream degradation processes. Two muscle-specific proteins, such as RING-finger protein-1 (MuRF1), an E3 ubiquitin ligase, and atrogin-1, an F-box protein belonging to a supramolecular complex that works as a ubiquitin ligase [39], are often found to be upregulated in *mdx* mice [31]. In addition, the upregulation of the E3 ubiquitin ligase tripartite motif-containing protein 32 (TRIM32) was observed in muscle tissues from patients affected by DMD or BMD [40,41]

These observations provide a rationale for an alternative therapeutic approach based on the inhibition of muscle-specific ubiquitin-conjugating enzymes that mark aberrant muscle proteins for proteasome degradation without impacting the entire proteasome. Recent studies have further identified four putative E3 ubiquitin ligases that selectively address missense variants of dystrophin as highly specific targets for chemical inhibition [42].

It has been proposed that the inhibition of these enzymes could be beneficial to diseased muscle tissues by preventing the ubiquitination and degradation of mutated dystrophin, thus increasing its cellular levels [40,42]. The selective ubiquitination of dystrophin can also be inhibited by using molecules that mimic long noncoding RNAs. One example is H19, a long noncoding RNA that interacts with the C-terminus of dystrophin, thus competing with TRIM63, a ubiquitin ligase enzyme expressed in both skeletal muscle and the heart, to bind to the same recognition site. The interaction with H19 inhibits the TRIM63-mediated polyubiquitination of dystrophin, thereby preventing its proteasome-mediated degradation [43]. The dystrophin mutant carrying the pathological amino acid substitution C3340Y is recognized with higher affinity by TRIM63, and therefore, it is subjected to over-ubiquitination and proteasome degradation to a greater extent than its wild-type counterpart [44]. When mice carrying the C3340Y mutation of dystrophin were treated with a chimeric, stabilized form of H19, an increased expression level of the mutant dystrophin was observed, which improved the morphology and function of skeletal and cardiac muscle [43]. Similar promising results have also been achieved through the administration of chimeric H19 to iPSC-derived skeletal muscle cells from BMD patients and *mdx* mice, who were pre-treated with drugs capable of inducing exon skipping to partially restore the amino acid sequence of dystrophin [43]. An improvement in the dystrophic phenotype was also observed with Nifenazone, which specifically targets TRIM63. This suggests that inhibiting the polyubiquitination-driven degradation of dystrophin, even in its mutated form, may yield beneficial effects [43].

#### 2.1.2. The Inhibition of the Immunoproteasome

As already mentioned, the partial or total loss of dystrophin found in DMD destabilizes the DGC, causing sarcolemma fragility and the activation of pro-inflammatory pathways, eventually leading to an increase in pro-inflammatory cytokines [45,46]. Among them, IFN-γ promotes the expression of three specific catalytically active β-subunits of the 20S proteasome, namely β1i (PSMB9), β2i (PSMB10), and β5i (PSMB8), which transform the constitutive proteasome normally expressed in healthy myofibers into a multi-subunit complex called immunoproteasome [47,48]. Such a complex is responsible for the production of peptides presented by the class I major histocompatibility complexes (MHC-I) to the antigen receptors of cytotoxic T cells [49]. It is therefore considered a therapeutic target in the context of inflammation and fibrosis. Despite not being directly responsible for muscle loss, the immunoproteasome stimulates signaling pathways that increase proteolytic degradation in an animal model of sarcopenia [50]. Since a degree of immunoproteasome upregulation was identified in the skeletal and cardiac muscle of *mdx* mice, the possibility was explored that its inhibition may hinder inflammation and rescue normal proteolytic degradation processes. Indeed, the treatment of *mdx* mice with ONX-0914, a specific inhibitor of the immunoproteasome, ameliorated the dystrophic phenotype and cardiomyopathy by reducing inflammation, oxidative stress, and fibrosis [51,52,53,54].

#### 2.1.3. The Limits of a Therapeutic Approach Based on the Inhibition of Proteasome-Driven Protein Degradation

Suppressing the degradation of a mutated protein by employing non-specific or highly specific inhibitors of proteasome-driven protein degradation represents a potentially risky strategy. In fact, the restoration of the expression levels of mutant proteins could be detrimental if they elicit aberrant functions. In addition, there are some contexts in which the inhibition of proteasome-driven protein degradation has not produced any benefit, since the low expression levels of dystrophin are independent of proteasome activity [55]. Interestingly, a study carried out on immortalized and undifferentiated myoblast cells from DMD patients, harboring mutations leading to prematurely truncated dystrophin, showed an accumulation of ubiquitinated protein aggregates containing HpsB5, a chaperone, which is known to protect key elements for muscle contractile function [56,57]. This was ascribed to reduced chymotrypsin-like activity of proteasome and an impaired capability of the autophagosome to fuse with lysosome for degradation [58].

To explain why proteasome inhibition does not always improve pathological conditions, it is essential to consider the wide genetic variability of DMD. Indeed, DMD patients often display large deletions involving more than one exon, but there is also a significant incidence of cases that are characterized by different defects of the dystrophin gene, including duplications of more than one exon, nonsense mutations, smaller deletions or insertions, splice site mutations, mid-intronic mutations, and missense mutations [59,60]. Different defects hitting the dystrophin gene have different repercussions on the expression levels and stability of dystrophin as well as on other members of the DGC. For instance, some dystrophin variants are misfolded but do not display any propensity to form insoluble aggregates inside the cells [42,61], while other dystrophin mutants more easily form large aggregates [58]. In the first case, it is very likely that aberrant proteins will be recruited for proteasome-mediated degradation, whereas the autophagy process (see below) will be activated for the clearance of large protein aggregates. Therefore, depending on the specific dystrophin defects, different cellular metabolic pathways could be activated, justifying the necessity for personalized therapeutic approaches.

### 2.2. Defective Autophagy and Signaling in DMD

Macroautophagy (hereafter referred to as autophagy) is a crucial catabolic intracellular process by which damaged or unnecessary cellular components, including nucleic acids, proteins, lipids, and organelles, are sequestered inside specialized vesicles called autophagosomes for recycling through lysosomal degradation.

Defective autophagy has been found in different contexts of muscle atrophy and disease [34]. A clue that autophagy was defective in DMD was provided as early as in 2011, when Eghtesad and colleagues showed that the administration of rapamycin, an inhibitor of mammalian target of rapamycin (mTOR) and a stimulator of autophagy, ameliorated the dystrophic phenotype in *mdx* mice by reducing the infiltration of T cells within the muscle fibers and the ensuing necrosis [62]. Indeed, in *mdx* mice and in DMD patients, autophagy was found to be impaired, although not equally in all muscles; for instance, controversial results have been obtained on the efficiency of autophagy in diaphragm [63,64]. Mitophagy is a process that selectively disrupts damaged mitochondria by autophagy. The sarcolemma fragility characteristic of DMD exposes muscular fibers to dysregulated intracellular Ca^2+^ fluxes and oxidative stress, with detrimental effects on mitochondria [65,66,67]. Despite the presence of dysfunctional mitochondria, mitophagy was found to be impaired in DMD animal models and human patients, leading to the loss of energy homeostasis and increased oxidative stress, contributing to disease progression [68,69]. Restoring autophagy, through the administration of rapamycin-loaded nanoparticles [70] or by feeding animals with a low-protein diet [63], reduced inflammation and fibrosis, producing consistent improvement in muscle function and cardiac contractile strength. The same happened when restoring mitophagy by administration of resveratrol [71] or urolithin [69] or by inducing the overexpression of the tripartite motif family protein (TRIM72), a protective myokine in *mdx* mice [72]. It was also hypothesized that ER-phagy—a selective form of autophagy involving the lysosomal degradation of specific subdomains of the endoplasmic reticulum—may be impaired in DMD. This hypothesis is supported by a recent study revealing the essential role played by ER-phagy for proper sarcoplasmic reticulum maturation during myoblast differentiation [73]

Although the autophagic defects in DMD are well known, the link between disease progression and impaired autophagic flux is not fully understood. To better clarify this aspect, the main anomalies observed in some signal transduction pathways and the way this affects autophagy in DMD are reviewed in the following paragraphs.

#### 2.2.1. Akt/mTOR Signaling

The Akt signaling cascade is one of the main pathways involved in protein synthesis and autophagy repression. By analyzing the diaphragm of *mdx* mice, Dogra and colleagues found the hyperactivation of phosphatidylinositol-3-kinase (PI3K) and Akt with a concomitant phosphorylation increase in their downstream targets, such as glycogen synthase kinase 3beta (GSK3β), forkhead box O1 (FoxO1), and mTOR. Interestingly, these authors measured further increased activation of PI3K and Akt compared to healthy animals when the diaphragm was subjected to passive mechanical stretch [74]. Further studies confirmed the upregulation of Akt activity in muscles other than the diaphragm, not only in *mdx* mice, but also in DMD patients [63,70,75].

Since PI3K/Akt signaling is known to promote muscle regeneration and hypertrophy, it is very likely that the increased Akt activity found in the dystrophic muscle represents an attempt to restore muscle integrity, inducing a state of proliferation/regeneration [76,77]. Indeed, in *mdx* mice, the upregulation of Akt activity was accompanied by increased levels of the dystrophin autosomal paralogue utrophin, which may be capable of replacing dystrophin when the latter is absent or lost [78]. While Akt activation plays a crucial role in promoting muscle growth and protein synthesis through the activation of mTOR complex 1 (mTORC1) [76,77], its persistent stimulation can have detrimental effects. Indeed, mTORC1 activation enhances protein synthesis but simultaneously inhibits autophagy [79,80], a process essential for cellular maintenance and homeostasis. Akt also phosphorylates and inhibits GSK3β, which would otherwise promote autophagy [81,82]. Additionally, Akt activation downregulates key autophagy-related genes, such as *Map1lc3b*, *Atg12*, *Gabarapl1*, and *Bnip3*. Pharmacological interventions that either inhibit mTOR, such as rapamycin and its analogs, or mimic caloric restriction by stimulating the fuel-sensing AMP-activated protein kinase (AMPK)—a key enzyme in energy homeostasis known to promote autophagy [83,84]—have been shown to restore autophagic flux and produce beneficial effects in *mdx* mice [63,64,85]. Indeed, the activation of AMPK was found to be reduced in the muscles of *mdx* mice [86,87].

Among the Akt/mTOR targets, there are FoxOs and TFEB, two transcription factors that foster autophagy by stimulating autophagy-related gene transcription [88,89]. It was reported that Akt phosphorylates FoxOs, driving their nuclear export, thereby preventing their transcriptional activity [90,91]. Similarly, the transcriptional function of TFEB is also regulated by its phosphorylation status. The phosphorylation of TFEB by mTORC1 triggers its nuclear export and represses its transcriptional activity [92,93,94]. As expected, the nuclear localization of FoxOs and TFEB was reduced, and their phosphorylation degree was increased in the skeletal muscle of *mdx* mice compared to healthy mice [64,95]. This was accompanied by the downregulation of many autophagy-related genes in the tibialis anterior and soleus muscles of 22-week-old *mdx* mice [95]. Downregulation concerned genes involved in autophagosome formation, such as *Atg5*, *Sqstm1*, *Map1lc3b*, and *Becn1*, and in the fusion process of autophagosomes and lysosomes, including *Rab7*, *Vps33a*, *Uvrag*, *Stx17*, *Vamp8*, *Snap29*, and *Ykt6*. Additionally, genes that participate in mitophagy, such as *Pink1*, *Bnip3*, and *Fundc1*, were also found to be downregulated in *mdx* mice. Among these genes, *Atg5*, *Sqstm1*, *Map1lc3b*, *Becn1*, *Rab7*, *Pink1*, and *Bnip3* are known to be FoxOs targets, whereas *Sqstm1*, *Map1lc3b*, *Becn1*, *Rab7*, *Vps33a*, and *Uvrag* are TFEB targets. Only the transcriptional levels of two other targets of TFEB, *Lamp1* and *Atp6v1a*, were not changed in *mdx* mice compared to healthy mice [95]. Interestingly, all of these genes display a very similar expression pattern in DMD patients, as evidenced by a meta-analysis carried out on data retrieved from the NCBI database [95]. These results strongly support the hypothesis that the transcriptional activation of autophagy-related genes is hindered by the reduced activity of FoxOs and TFEB in *mdx* mice as well as human patients [95].

Although the hyperactivation of Akt and mTORC1 may be a plausible mechanism to suppress autophagy in DMD, there are still many unclear aspects that warrant further investigations. For instance, some studies were carried out on *mdx* mice, where Akt activity was unchanged compared to healthy animals [96]. Accordingly, a recent report proposed that the hyperactivation of mTORC1 found in *mdx* mice is not due to the dysregulation of the Akt activity but to the overexpression of leucyl-tRNA synthetase. The latter activates mTORC1 via a noncanonical mechanism based on the activation of RagD, a GTPase that mediates the recruitment of mTORC1 to the lysosome surface [97]. In addition, beneficial effects have been observed in *mdx* mice by suppressing the gene encoding for the serum- and glucocorticoid-inducible kinase (SGK), which phosphorylates FoxOs at the same sites as Akt [98]. Therefore, the possibility that SGK or other kinases may also be responsible for the phosphorylation of FoxOs cannot be ruled out. Sirtuin 1 (SIRT1) could also influence the intracellular localization of FoxOs; in fact, SIRT1 is known to deacetylate FoxOs [99] and to drive their nuclear localization [100,101]. The treatment of *mdx* mice with resveratrol, a known activator of SIRT1, restored mitochondrial autophagy through the upregulation of FoxO-target genes, probably due to the FoxOs increased nuclear levels [71,102].

However, there are some circumstances where the upregulation of PI3K/Akt signaling is not linked to autophagy downregulation. For example, the treatment of *mdx* mice with aminoguanidine hemisulfate, a known inhibitor of nitric oxide synthase and reactive oxygen species (ROS), ameliorated both the histological phenotype and muscle function by restoring mitochondrial autophagy. Nevertheless, this was associated with a further increase in the phosphorylation of Akt and FoxO1 that was already enhanced in *mdx* mice in comparison to healthy controls [103].

A recently described plakoglobin-dependent interaction between the DGG and the insulin receptor has been proposed to play a key role in supporting insulin-induced PI3K/Ak signaling in murine skeletal muscle [104]. In addition, it has been shown that in rabbit skeletal muscle, the G_βγ_ dimer associated with the DGC recruits and activates PI3K, which in turn activates Akt. This is strictly dependent on the binding of laminin to α-dystroglycan, as the disruption of this interaction impairs PI3K/Akt signaling [105,106]. The absence of dystrophin likely weakens the binding between α-dystroglycan and laminin as well as the interaction between the DGC and the insulin receptor, which would normally lead to the downregulation of the PI3K/Akt signaling pathway and to an increase in autophagic flux. However, this outcome contrasts with other experimental observations. Interestingly, the loss of dystrophin and other components of the DGC triggers a compensatory upregulation of integrin α7β1 in both *mdx* mice and DMD patients [107,108,109]. Moreover, integrin binding to laminin can stimulate cell signaling in a manner similar to the mechanism observed for α-dystroglycan [110,111,112]. Therefore, it is plausible that in pathological conditions, integrin α7β1 may overactivate PI3K/Akt signaling to compensate for the loss of DGC [105,106] (Figure 3).

#### 2.2.2. Hyppo Pathway

Besides the Akt/mTOR axis overactivation, other aberrant events influencing signal transduction pathways can be invoked to explain the autophagy impairment observed in DMD. The Hippo pathway is a kinase cascade that culminates in the phosphorylation of two transcriptional co-factors, Yes-associated protein 1 (YAP) and transcriptional coactivator with PDZ-binding motif (TAZ), which leads to their retention in the cytoplasm and proteasomal degradation [113]. Conversely, when the Hippo pathway is inactive, unphosphorylated YAP/TAZ translocate to the nucleus, where they serve a role as coactivators of the transcriptional enhancer factor (TEA)-domain (TEAD1–4) family of transcription factors, stimulating the transcription of their target genes and inducing cell proliferation.

The Hippo pathway plays a crucial role in transducing mechano-sensing signals that regulate cellular behavior. The physical properties of the extracellular matrix profoundly influence key cellular structural features, such as polarity, shape, and cytoskeletal organization, which in turn modulate YAP/TAZ activity. Additionally, various mechanical and environmental stimuli, including matrix stiffness, energetic stress, osmotic stress, and hypoxia, can further regulate the Hippo pathway, highlighting its central role in integrating mechanical cues [114]. In turn, YAP/TAZ activity regulates a plethora of cellular functions, including actomyosin contractility and the organization of the actin filaments [115,116]. YAP also restores skeletal muscle mass and protein synthesis following injury and atrophy [117]. In healthy mice, contractile loading leads to nuclear YAP activity, while, in *mdx* mice, YAP is constitutively activated and does not respond to contractile loading [118]. In contrast, the extent of YAP phosphorylation has been found to be increased in muscle biopsies from DMD patients with respect to controls, suggesting a downregulation of YAP/TAZ nuclear activity [119,120]. Suppression of the nuclear activity of YAP has also been demonstrated in myoblasts expressing dystrophin variants carrying missense mutations linked to DMD or BMD [121]. Accordingly, a phospho-transcriptome analysis carried out on neural stem cells derived from DMD patients displaying intellectual disability revealed an altered expression of genes related to the Hippo signaling pathway with a concomitant hyper-phosphorylation of YAP [121,122]. In non-muscle cells and tissues, inhibition of the transcriptional activity of YAP/TAZ blocks autophagy by suppressing the transcription of *Armus*, one of their target genes [123]. Indeed, Armus, which is a RAB7-GTPase activating protein, interferes with the fusion of autophagosomes with lysosomes [124,125]. By analogy, this would suggest that the downregulation of YAP/TAZ activity might contribute to impaired autophagy in DMD.

Remarkably, it has been found that β-dystroglycan interacts directly with YAP through its PPYP motif, which is also responsible for the interaction with the WW domain of dystrophin or utrophin, and that these interactions can be modulated by different α-dystroglycan extracellular binding partners [126,127,128]. For instance, in mice neonatal heart, agrin displays high expression levels and forms a stable complex with α-dystroglycan. This elicits a stabilizing effect on the interaction between β-dystroglycan and dystrophin. In the heart of adult mice, the expression levels of agrin drop down, leading α-dystroglycan to preferentially bind to other still unidentified extracellular proteins [129]. This is hypothesized to induce conformational changes within the dystroglycan complex, ensuing the release of dystrophin in favor of YAP. The sequestration of YAP by β-dystroglycan prevents this transcriptional factor from eliciting its function in the nucleus, where it induces cell growth and proliferation [127]. Since dystrophin and YAP compete in binding to the same PPYP motif within the C-terminus of β-dystroglycan, it could be speculated that in DMD, the loss of dystrophin could excessively reinforce the binding between β-dystroglycan and YAP, impairing the nuclear activity of YAP and the cellular functions that arise from it, including autophagy (Figure 4). It is worth noting that in DMD, the β-dystroglycan C-terminus, containing the PPYP binding epitope, responsible for the interaction with different binding partners within the cytosol, remains intact even when β-dystroglycan is exposed to the activity of metalloproteinases, such as MMP-2 and MMP-9. Indeed, many studies carried out on animal models of DMD highlighted the presence of a 30 kDa truncated form of β-dystroglycan, spanning the full-length protein, devoid of its N-terminal extracellular domain [130].

#### 2.2.3. RhoA/ROCK Signaling

The small GTPase RhoA and its major effector Rho-associated protein kinase (ROCK) are key elements of a signal transduction pathway that modulates many cellular functions, including actin cytoskeleton reorganization [131]. Many studies have proven that RhoA/ROCK signaling inhibits autophagy, although the molecular mechanisms underlying these effects are still not completely clarified [132]. For example, it has been suggested that RhoA/ROCK signaling can modulate the autophagosome formation [133] or can interfere with the Akt/mTOR axis [134]. Moreover, it has been proposed that RhoA/ROCK signaling might modulate trafficking from early autophagosomes to late autolysosomes [135,136]. Recently, increased expression levels of ARHGEF3, a RhoA guanine nucleotide exchange factor, observed in the muscles of *mdx* mice, have been found to elicit RhoA/ROCK signaling upregulation [137]. Of note, the deletion of *Arhgef3* in *mdx* mice restored muscle strength and morphology through the rescuing of autophagy flux [137]. Other studies confirmed the persistent activation of RhoA/ROCK signaling in the skeletal muscle of *mdx* mice. Its inhibition stimulated the muscle differentiation of myoblast cells devoid of dystrophin and its homolog utrophin and increased myofiber quantity, ameliorating the phenotype [138,139].

All of this evidence indicates an interplay between the DGC and RhoA/ROCK signaling, although—to the best of our knowledge—direct protein interactions between members of the DGC and proteins participating in RhoA/ROCK signaling have never been identified. It cannot be ruled out that the dysregulation of RhoA/ROCK signaling observed in DMD may be at least in part due to the destabilization of the DGC, thus contributing to the impaired autophagy. This hypothesis is supported by the notion that β-dystroglycan recruits, in a laminin-dependent way, the growth factor receptor-bound protein 2 (Grb2), an adaptor protein involved in signal transduction and cytoskeleton organization [140]. In turn, Grb2 recruits other proteins belonging to the Rho GTPase family, such as Rac1, Cdc42, H-Ras, and RhoA, to sustain cell survival [141,142] (Figure 5). It is worth mentioning that RhoA, Rac1, and Cdc42 also contribute to the shape of the actin cytoskeleton by involving numerous downstream effector proteins [143], and this may have important repercussions on autophagy (see below and Figure 5). In pathological conditions, the RhoA/ROCK axis could synergize with the Hyppo pathway to repress autophagy since it has been proven that RhoA activation inhibits YAP in muscle satellite cells [144].

#### 2.2.4. MEK/ERK Signaling

The mitogen-activated protein kinase (MAPK) signaling cascades that modulate cell proliferation, differentiation, and survival were found to be aberrant in *mdx* mice. Indeed, dystrophic animals showed increased phosphorylation levels of the extracellular signal regulated kinases 1 (ERK1) and 2 (ERK2) [145,146,147] and of the c-jun N-terminal kinases 1 (JNK1) and 2 (JNK2) [148,149,150], as well as reduced phosphorylation of p38 [147]. In physiological conditions, MEK and activated ERK directly interact with β-dystroglycan through its juxtamembrane region; it was proposed that β-dystroglycan may modulate the MEK/ERK pathway as it recruits MEK within the membrane ruffles, hindering MEK-driven ERK phosphorylation [151]. No direct link between the upregulation of the MEK/ERK pathway and the blockage of autophagy in DMD has yet been characterized. Moreover, the role of MEK/ERK in the regulation of autophagy is quite complex to define, and different studies produced controversial results [152,153,154,155,156,157].

#### 2.2.5. The Actin Cytoskeleton Scaffold

A link between the actin cytoskeleton and autophagy emerged as early as in 1992, when Aplin and colleagues found that cells deprived of nutrients and treated with actin-disrupting compounds like cytochalasin D and latrunculin B could not form autophagosomes [158]. Later studies showed that actin filaments co-localize with key autophagy markers [159,160]. Moreover, a mouse model lacking the *Atg7* gene, which plays a key role in autophagy, is characterized by both impaired autophagosome formation and defective actin assembly, suggesting a probable interconnection between these two cellular processes [161].

Actin polymerization is a highly reversible process, which is finely tuned by many actin-binding proteins [162]. Monomeric actin, called G-actin, may polymerize to produce actin filaments called F-actin. Recent studies indicated that excessive stabilization of F-actin impairs autophagic activity in a *Drosophila* Parkinson’s disease model characterized by α-synuclein neurotoxicity. Indeed, the destabilization of F-actin obtained by overexpressing actin filament-severing proteins, like gelsolin or cofilin, restored autophagic flux and facilitated the removal of α-synuclein aggregates and dysfunctional mitochondria [163]. Similarly, excessive F-actin polymerization, found in the aged brain of *Drosophila*, impairs autophagic activity and causes the accumulation of dysfunctional mitochondria. The treatment of aged *Drosophila* with drugs that disrupt actin polymerization rescued normal brain autophagic flux and improved cognitive performance [164]. The hypothesis that excessive F-actin polymerization may also occur in dystrophic muscles is supported by the observation that microtubule-associated monooxygenase 2 (MICAL2), a protein that stabilizes F-actin, by modulating its polymerization and depolymerization, was found to be upregulated in muscles of *mdx* mice [165].

What is now clear is that actin plays key roles in every stage of autophagy [166]. Indeed, actin filament networks provide structural support during the generation of a double-membrane structure, called phagophore, which encloses the damaged material and preludes the formation of the autophagosome through the fusion of its extremities [167,168]; furthermore, actin dynamics regulate autophagosome trafficking to the lysosome [169] and the fusion of autophagosomes with lysosomes, where the degradation of the macromolecules that must be recycled occurs [170,171].

Actin polymerization harnesses many nucleator factors, some of which are involved in autophagy [166,172]. For example, several lines of evidence highlight the critical role of the Arp2/3 complex in actin polymerization during different stages of autophagy [166,167,173,174,175,176]. The Arp2/3 complex participates in the early stages of actin polymerization by promoting the generation of branched actin networks, which support many membrane-remodeling processes, including autophagy [166,177]. As a matter of fact, the inhibition of the Arp2/3 complex’s activity prevents the formation of the omegasome, a specialized membrane structure that the autophagosome originates from [167,173,178,179]. When the autophagy process is activated, the actin nucleator WHAMM relocates to the omegasome by a still unknown mechanism and recruits the Arp2/3 complex [178]. Branched actin networks, associated with the Arp2/3 complex, polymerize inside the developing phagophore [167]. This provides structural architecture for supporting membrane curvature during phagophore expansion, thus allowing for the fusion of the ends to enclose the autophagosome [168].

The actin cytoskeleton also serves an important function during the fusion of autophagosomes with lysosomes [171]. In selective autophagy, cortactin, an actin remodeling factor, localizes to autophagosomes, where it recruits the Arp2/3 complex to promote actin polymerization and the rearrangement of actin filaments. The assembly of an F-actin network facilitates the fusion of autophagosomes with lysosomes for the subsequent degradation of the material entrapped within the autophagosome [171]. There is also some evidence that actin polymerization supports the trafficking of autophagosomes during starvation-induced autophagy [178,180].

An impaired autophagosome/lysosome fusion has been observed in the skeletal muscles of *mdx* mice, which also exhibit a reduced lysosomal abundance. Furthermore, there is a notable increase in the expulsion of autophagosomes from muscle fibers into the extracellular matrix, which is significantly more pronounced in *mdx* mice than in healthy controls [87].

Many lines of evidence suggest that the DGC can also translate mechanical stimuli from the extracellular matrix into biochemical signals inside the cells, a process called mechanotransduction, thus regulating cytoskeleton plasticity [10]. As a matter of fact, dystrophin establishes direct interactions with both G- and F-actin through its N-terminus and its central rod domain [181,182,183], thus contributing to modulate cytoskeletal architecture and dynamics by maintaining G-actin and stabilizing microtubule lattice formation at the sub-sarcolemma [181,184]. The absence of dystrophin in *mdx* mice has been shown to lead to significant changes in the cytoskeletal structure, affecting a variety of proteins and disrupting the overall organization of the cytoskeleton [185]. Indeed, increased G-actin levels associated with the subsarcolemmal lattice were observed in *mdx* mice as compared to healthy mice, whereas the link between the sarcolemma and F-actin further inside the cell was interrupted [186].

Other members of the DGC contribute to mechanotransduction; for example, in fibroblasts, cell adhesion promotes β-dystroglycan recruitment of the peripheral membrane protein ezrin, a cytoskeletal adaptor that enables β-dystroglycan to interact with F-actin. The formation of a ternary complex constituted by the Rho GTPase protein Dbl, β-dystroglycan, and ezrin primes a signal transduction pathway involving Cdc42 that culminates in the formation of filopodia and microvilli [187,188,189]. Another example is the filopodia of non-muscle cells, where the interaction of α-dystroglycan with laminin has been shown to modulate and reinforce β-dystroglycan clustering with F-actin, promoting the cytoskeleton remodeling required for correct filopodia development and cell migration [190]. In addition, the discovery of a complex composed of β-dystroglycan, Src kinase, and tyrosine kinase substrate 5 (Tks-5) in myoblasts supports the hypothesis of a possible involvement of dystroglycan in the formation of podosomes [191]. The latter represent actin-rich structures that sustain various cell processes, such as migration and morphogenesis.

Although autophagosomes, microvilli, filopodia, and podosomes perform distinct functions within the cell, the molecular mechanisms underlying the generation of these actin-rich structures share some characteristics, including membrane dynamics and actin cytoskeleton remodeling. While the formation of microvilli, filopodia, and podosomes requires local extension of the plasma membrane, autophagosomal membranes originate from different sources besides the plasma membrane, including the endoplasmic reticulum and Golgi apparatus. Nevertheless, the formation of all of these structures needs deep actin cytoskeletal reorganization that involves many actin-binding proteins.

The aberrant network of protein–protein interactions, induced by the loss of dystrophin and other members of the DGC, causes a deregulation of actin dynamics that could have important repercussions on the different autophagic steps (Figure 6). This hypothesis is further supported by a recent analysis of the β-dystroglycan interactome carried out on Hek-293 cells expressing only the β-dystroglycan subunit, which identified a relevant number of actin binding proteins. Among them, there is the actin-related protein 3 (Arp3), belonging to the Arp2/3 complex, as well as talin, vinculin, alpha-actinin-4, and the already mentioned ezrin that could establish a protein network with dystroglycan and affect cytoskeleton reorganization [192].

## 3. Dystroglycanopathies Associated with Fukutin and Fukutin-Related Protein

A heterogeneous group of congenital muscular dystrophies, also called dystroglycanopathies, are associated with anomalies of the genes encoding for at least 20 enzymes required for the multi-step glycosylation of α-dystroglycan [193,194,195]. The complex polysaccharide structure, called *matriglycan*, covalently linked to α-dystroglycan, is responsible for mediating its binding to laminin and other extracellular proteins containing laminin-globular (LG) domains [196,197,198,199,200]. The incorrect glycosylation of α-dystroglycan prevents it from binding to laminin, thus interrupting the connection between the extracellular matrix and the cytoskeleton. Although the disruption of the extracellular matrix/cytoskeleton axis is a common hallmark of all known dystroglycanopathies, these conditions exhibit a broad spectrum of neuromuscular symptoms. These range from mild clinical manifestations in limb–girdle muscular dystrophy type R9 (LGMDR9, previously known as LGMD2I) to more severe presentations in congenital muscular dystrophies, including congenital muscular dystrophy type 1C (MDC1C), Walker–Warburg syndrome (WWS), and muscle–eye–brain disease (MEB), which are also associated with cognitive impairments and significant brain defects [12,194,201].

Congenital muscular dystrophies linked to fukutin and fukutin-related proteins constitute two distinct subgroups of dystroglycanopathies, for which comprehensive analyses of proteasomal activity and autophagic flux have been carried out. Fukutin (FKTN) and its paralogue fukutin-related protein (FKRP) are two ribitol-5-phosphate transferases located in the Golgi apparatus, required for the sequential ribitol phosphate modifications of the α-dystroglycan *matriglycan*. Mutations hitting the genes encoding FKTN and FKRP cause different forms of congenital muscular dystrophies [202,203]. Genetic variants of the *FKTN* and *FKRP* genes are associated with a wide spectrum of phenotypes ranging from mild to severe clinical signs, including cognitive impairment [202,204].

### 3.1. The Upregulation of the Ubiquitin–Proteasome System

Muscle biopsies obtained from LGMDR9 patients harboring the homozygous missense variant *FKRP-L276I* showed elevated global levels of protein ubiquitination coupled with the upregulation of the α5 (PSMA5) and β5 (PSMB5) subunits of the 20S catalytic core of the 26S proteasome. In addition, enhanced expression levels of two key markers of endoplasmic reticulum stress, protein disulfide isomerase (PDI) and phosphorylated eukaryotic translation initiation factor 2α (peIF2α-Ser51), were observed. These findings indicate that both heightened proteasome activity and ER stress play a role in disease progression [205]. Nevertheless, the protein expression levels of the two E3 ubiquitin ligases enzymes, atrogin-1 and MuRF1, which are known markers of muscle atrophy [39,206], were not changed compared to the control samples [207]. Notably, in C2C12 murine muscle cells, the FKRP-L276I variant, which is associated with the mild LGMDR9 [202], correctly localized within the Golgi apparatus, like its wild-type counterpart, while other variants associated with more severe forms of congenital muscle dystrophies, such as MDC1C [201,207], were found entrapped within the endoplasmic reticulum, where they can be removed by the endoplasmic reticulum-associated degradation (ERAD) pathway involving the proteasome [208,209]. This may suggest that the *FKRP-L276I* mutation can result in a more severe phenotype in humans compared to mice, as often observed in murine models that fail to recapitulate dystrophic phenotypes [210,211].

### 3.2. Defective Autophagy

Studies evaluating autophagic flux in the context of defective *FKTN* or *FKRP* genes suggest a dysregulated autophagic process, though the findings point to opposing directions [205,212,213].

#### 3.2.1. Akt/mTOR Signaling

Enhanced activation of mTOR kinase, but not Akt, was observed in a conditional knockout mouse model of LGMDR9 in which the *FKTN* gene was deleted during muscle development. This led to a range of dystroglycanopathies, from mild to severe, mirroring the phenotypic variability seen in human patients [212]. Interestingly, the phosphorylated form of S6 ribosomal protein, which is a downstream target of mTORC1, was found to colocalize with pathological markers of fibrosis and acute regeneration, indicating a direct correlation between the hyperactivation of mTORC1 and muscular dystrophy [212]. The elevation of specific autophagic markers, such as beclin-1, has been interpreted as evidence of increased autophagic flux due to the deletion of the *FKTN* gene, although the administration of rapamycin, which is known to stimulate autophagy by inhibiting mTORC1, ameliorated the histological features of skeletal muscle [212]. Remarkably, in the presence of α-dystroglycan hypoglycosylation driven by an *FKTN* disruption, an analysis of the phosphorylation status of Akt, mTOR kinase, and S6, conducted before the onset of muscular dystrophy, revealed no significant differences compared to wild-type mice. This suggests that mTORC1 hyperactivation is not directly attributable to α-dystroglycan hypoglycosylation [212].

A study carried out on muscle samples from patients affected by LGMDR9, expressing the homozygous missense variant FKRP-L276I, revealed that the phosphorylation status of Akt, which is a measure of its activation, was unchanged, whereas the extent of phosphorylation of mTOR (Ser2448) and GSK3β (Ser9), two downstream targets of Akt, was reduced with respect to the control [205]. The protein levels of Atg7 and LC3-II, which are involved in the early stages of autophagy, were found to be increased, whereas that of p62, which represents a cargo receptor and is itself degraded via autophagy, were reduced. This result has been interpreted as an indication of elevated autophagy, further corroborated by the increased co-localization of LC3-II, p62, and the lysosome-associated membrane protein 2 (LAMP-2) within muscle fibers [205]. However, no obvious correlations have been found among the autophagy markers analyzed and disease progression, except for LC3-II, which was inversely correlated with age at onset [205].

A transcriptome analysis on pluripotent stem cell-derived myotubes, in which the *FKRP* gene has been disrupted, showed opposite results, i.e., the downregulation of many genes involved in lysosome degradation was observed, indicative of an impaired autophagic flux [213]. A further analysis carried out on pluripotent stem cell-derived myotubes from human patients carrying different genetic variants of the *FKRP* gene, associated with congenital muscle dystrophy of different severities, including LGMDR9 and WWS, confirmed a blockage of autophagic flux in all the phenotypes analyzed without any involvement of the Akt/mTOR axis, which was unchanged compared to the control cells. Notably, a complete loss of α-dystroglycan glycosylation was found in cells from WWS patients, whereas the extent of α-dystroglycan glycosylation in cells from LGMDR9 patients was like that found in the control cells [213,214,215,216]. This result indicates that the blockage of autophagic flux is not related to the extent of α-dystroglycan glycosylation.

The partially conflicting results reported for the activation status of the Akt/mTOR signaling pathway may reflect the different models in which they were collected. When evaluating autophagic activity, it is essential to recognize that studies involving muscle tissues from *FKTN* knockout mice [204] and human patients with a single point mutation in the *FKRP* gene [197] have limitations. These studies take place in a static context, making it difficult to monitor the trafficking of cargo to lysosomes for degradation. In this setting, it is very hard to obtain conclusive results since any observed increase in autophagic markers could result from either the upregulation of autophagy or a blockage at the later stages of the autophagic flux.

#### 3.2.2. Potential Involvement of Other Signal Transduction Pathways

In addition to the Akt/mTOR pathway, other signaling pathways potentially involved in the regulation of autophagy have also been explored. For example, the activation state of promoters of autophagy, such as AMPK and tumor necrosis factor receptor-associated factor 6 (TRAF6), was not found to be different in the skeletal muscles of LGMDR9 patients with respect to their healthy counterparts [205].

In contrast, ERK1/2 kinase activity was found to be reduced in *FKRP* KO cells and in those from WWS patients, characterized by severe hypoglycosylation of α-dystroglycan. In contrast, cells from LGMDR9 patients displayed almost normal α-dystroglycan glycosylation compared to control cells [213]. This is coherent with the discovery of reduced ERK activity upon the disruption of α-dystroglycan/laminin binding [217]. Nevertheless, no correlation was found between ERK1/2 activity levels and autophagy, either in cells or in patients [213].

## 4. Sarcoglycanopathies

Sarcoglycanopathies are a group of autosomal recessive muscle disorders characterized by genetic alterations of four transmembrane glycoproteins within the DGC, called sarcoglycans [218]. In skeletal and cardiac muscle, sarcoglycans assemble in an inner core made of β-, γ-, and δ-sarcoglycans that interacts with α-sarcoglycan and dystroglycan, providing structural support to the sarcolemma [9]. Sarcoglycanopathies are considered a subgroup of LGMD as they mainly affect proximal muscles around the scapular and the pelvic girdles. Although the clinical symptoms of sarcoglycanopathies are extremely variable, they all lead to muscle weakness and atrophy, often causing premature death for respiratory or cardiac impairment [219]. Although a direct genotype/phenotype correlation has not yet been clearly defined, the genetic defects that result in the complete loss of one of the sarcoglycans are typically associated with the most severe forms of LGMD in contrast to mutations that cause only a reduction in the levels of one member of the sarcoglycan complex [220,221]. In any case, the entire sarcoglycan complex is destabilized, and this may have repercussions on the stability of the whole DGC and on the integrity of sarcolemma. Along their secretory pathway, sarcoglycans are scrutinized by a sophisticated quality control system, which marks mutated variants for retrotranslocation to the cytosol, where they are degraded by the proteasome [222,223,224,225]. However, some sarcoglycan mutants, although misfolded, may maintain their functionality and can be rescued by inhibiting proteasome activity [225,226]. In addition, drugs called cystic fibrosis transmembrane regulator (CFTR) correctors, which were originally developed to rescue a pathologic mutant of the CFTR, have proven to restore the normal protein levels of mutated sarcoglycans and their correct membrane localization in myotubes [227]. In mouse models of two different sarcoglycanopathies, these drugs have demonstrated efficacy in restoring the normal muscular phenotype and functionality [226,228].

## 5. Congenital Muscular Dystrophies Associated with Laminin α2 Deficiency

Mutations within the *LAMA2* gene, which encodes for the α2 subunit of the basement membrane protein laminin-211, i.e., the main isoform of muscle laminin, may cause the complete loss or a partial reduction in laminin, leading to severe or milder forms of congenital muscular dystrophies [229,230], respectively. The most severe form is the so-called congenital merosin-deficient muscular dystrophy type 1A (MDC1A) that is characterized by variability of the muscle fiber size and continuous cycles of degeneration/regeneration, accompanied with muscle weakness and atrophy that lead to death in the first decade of life for around 30% of patients [231]. The main symptoms of MDC1A are recapitulated by the *dy^3K^/dy^3K^* mouse, an animal model characterized by the complete ablation of the *LAMA2* gene.

### 5.1. The Upregulation of the Ubiquitin–Proteasome System

An analysis of muscle tissues of this animal model showed transcriptional upregulation of two members of the FoxOs transcription factors family, namely FoxO1 and FoxO3, and their targets atrogin-1 and MuRF1 [232]. Additionally, *dy^3K^/dy^3K^* mice exhibited a significant increase in global protein ubiquitination, which only became evident following the onset of dystrophic symptoms [232]. The treatment of *dy^3K^/dy^3K^* mice with proteasome inhibitors, such as MG-132 or Bortezomib, partially restored the normal activation status of Akt, improved muscle phenotype and locomotive activity, and doubled lifespan [232,233]. All together, these findings indicate that the upregulation of proteasome degrading activity contributes to disease progression.

### 5.2. Defective Autophagy

Mice lacking the *LAMA2* gene showed increased expression levels of many genes involved in autophagy when muscles displayed clear signs of dystrophy, whereas their levels were not different from their healthy counterparts in younger mice before the onset of dystrophic symptoms [234]. Furthermore, *dy^3K^/dy^3K^* mice showed reduced Akt phosphorylation compared to healthy littermates, in line with the results of previous research showing that the PI3K/Akt pathway is disrupted when the interaction between the DGC and laminin is lost [105,106]. Analogously, in myotubes, but not in myoblasts derived from patients affected by MCD1A, the expression levels of several autophagic markers were increased with respect to the control, suggesting the overactivation of autophagy that accompanies disease progression [234]. A more recent comparative study carried out on biopsies from newborns affected by MCD1A and other muscular dystrophies, including DMD, showed a similar accumulation of LC3-II in all of the muscles analyzed, whereas the accumulation of p62 could be observed, for example, in DMD but not in MCD1A muscles. Although this analysis was performed in static conditions, the comparison between DMD and MCD1A strongly suggests a blockage of autophagic flux in DMD but not in MCD1A, where autophagy was overactivated [235]. The administration of 3-methyladenine, a drug that inhibits autophagosome formation, to *LAMA2*-deficient mice partially restored muscle phenotype by reducing fibrosis, increasing the average fiber section and ameliorating muscle morphology, providing further demonstration that increased autophagic flux is strictly related to disease progression [234]. Interestingly, mice deficient in the *LAMA2* gene, overexpressing a DNA construct encoding a truncated form of the laminin α1 chain that is capable of binding to integrin but not dystroglycan, display no alteration of the expression levels of autophagy-related genes, indicating that dystroglycan is not directly involved in the autophagy regulation triggered by the lack of laminin α2 [234].

## 6. Conclusions

The loss of protein homeostasis in muscle is a common hallmark of the main muscular dystrophies associated with the DGC and significantly contributes to muscle waste and atrophy. In fact, extensive research in the field showed that a range of degradative actions driven by the upregulated ubiquitin–proteasome system and an alteration (mostly impairment) of the main autophagy flux disrupt the delicate balance between protein synthesis and degradation. While excessive proteasome-driven protein degradation is probably due to the accumulation of mutant misfolded proteins that could be dangerous for muscle cells, less clear are the reasons for the alteration in autophagic flux that is observed in pathological conditions. From a therapeutic standpoint, the inhibition of the whole ubiquitin–proteasome system has proven to be poorly effective if not deleterious, especially for long-term treatments, since it leads to the accumulation of misfolded and damaged proteins that could produce detrimental effects. Similarly, the highly specific inhibition of enzymes that selectively target misfolded or truncated dystrophin variants for degradation may cause exposure to risks due to possible toxic effects elicited by such dystrophin mutants.

Given that the DGC participates in multiple signal transduction pathways that regulate a wide range of cellular functions, including protein metabolism, our survey suggests the presence of potential pathological aberrations within the key signaling pathways involving the DGC, particularly those that impact autophagic flux. Moreover, many studies stress the importance of the interaction between the DGC and the actin cytoskeleton, with potential consequences of aberrant cytoskeletal remodeling on various stages of autophagic flux. From our survey, it could be foreseen that strategies aimed at strengthening cellular recycling, rather than inhibiting the proteasome/ubiquitin pathway, could represent promising therapeutic avenues for the treatment of muscular dystrophies.

## Figures and Tables

**Figure 1 cells-14-00721-f001:**
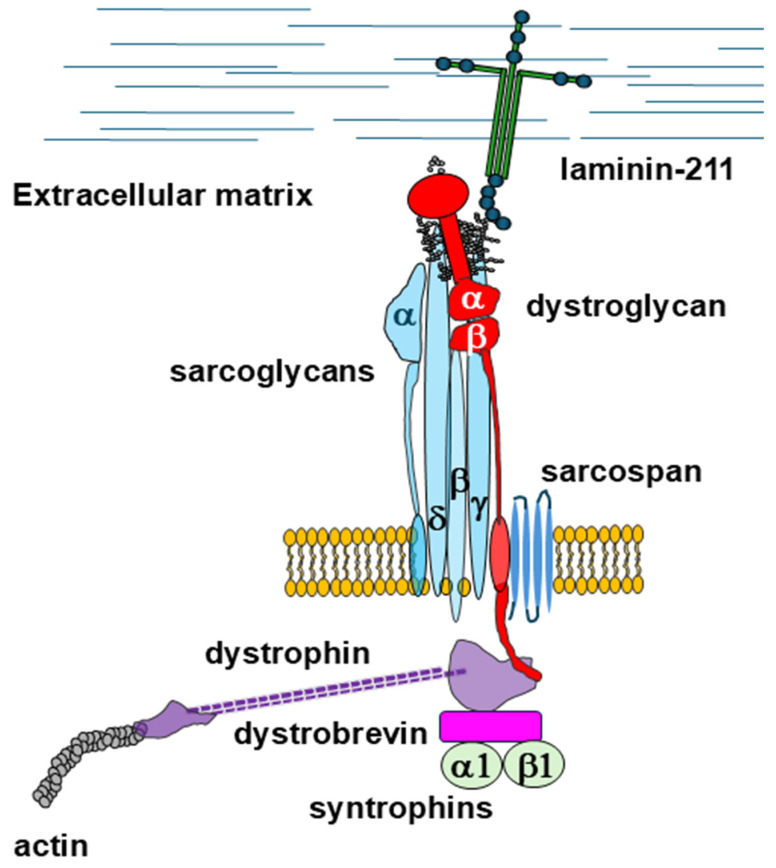
A schematic representation of the most important proteins that constitute the DGC. α-dystroglycan is heavily glycosylated and plays a crucial role within the whole complex as it harbors the glycosylated moieties (the so-called *matriglycan*) recognized by extracellular binding partners, such as laminins and others. Molecular recognition at the dystroglycan axis ensures a direct connection between the extracellular matrix and the cytoskeleton, with α-dystroglycan non-covalently binding to its transmembrane partner, β-dystroglycan, whose C-terminal domain binds directly to the dystrophin–actin cytoskeleton.

**Figure 2 cells-14-00721-f002:**
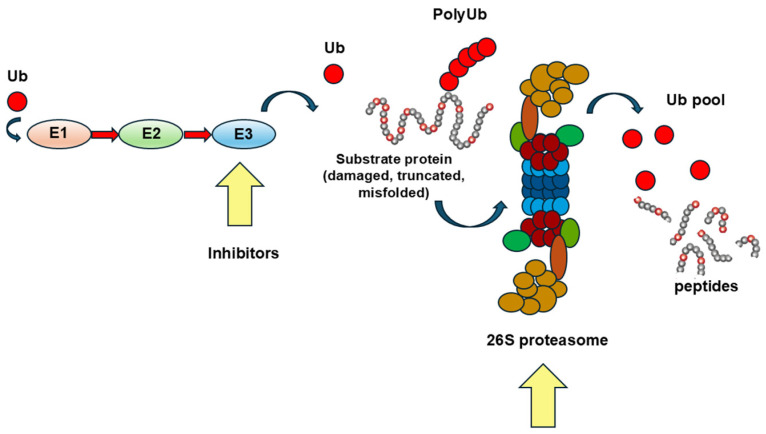
The ubiquitin–proteasome system (UPS) is composed of a multi-subunit enzymatic complex, whose function is degrading misfolded, truncated, or damaged proteins after they are recognized and bound by ubiquitin (a small 8.6 kDa regulatory protein). In muscular dystrophies, components of the DGC lose their proper transmembrane or juxtamembrane localization and can become unstable, making them prone to be degraded by the proteasome upon ubiquitination. Multiple enzymatic steps (E1, E2, and E3 cascades) are necessary to attach several ubiquitin (Ub) units to the substrate protein that is subsequently degraded in the 26S proteasome. The inhibition of the UPS at different steps (yellow arrows) is a therapeutic avenue aiming to limit the negative impact of degrading dystrophin, or other DGC proteins, on skeletal muscle stability.

**Figure 3 cells-14-00721-f003:**
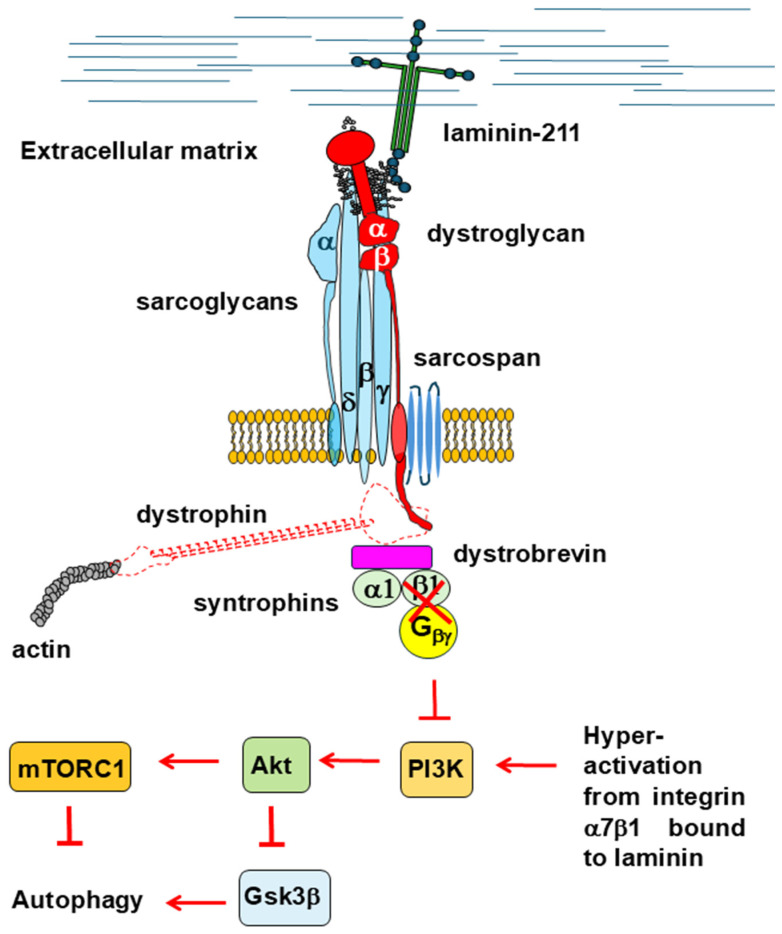
The loss of dystrophin disrupts the interaction that the DGC establishes with the beta/gamma subunits of the G protein. The DGC is thus no longer capable of modulating PI3K/Akt signaling. The compensatory increase in the integrin α7β1 expression levels found in these conditions probably overactivates the PI3K/Akt/mTORC1 axis, which promotes protein synthesis while inhibiting autophagy.

**Figure 4 cells-14-00721-f004:**
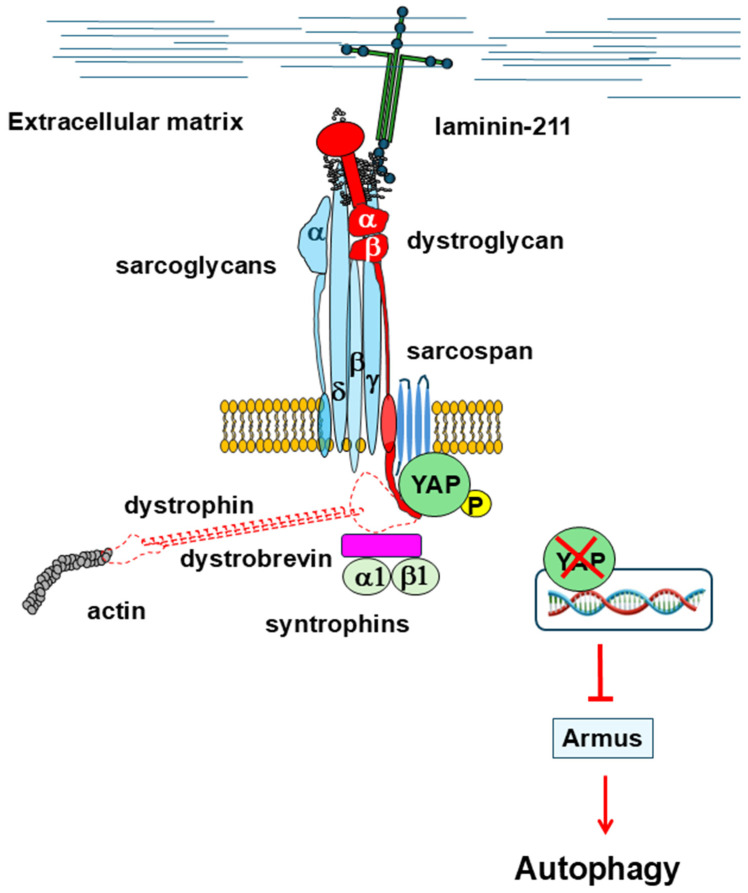
In the absence of dystrophin, β-dystroglycan could reinforce binding with the phosphorylated form of YAP, preventing its nuclear localization/activity and the consequent expression of its target genes, including *Armus*. The protein encoded by *Armus* plays a key role in the fusion of autophagosomes with lysosomes, and its loss likely hampers the normal autophagy flux.

**Figure 5 cells-14-00721-f005:**
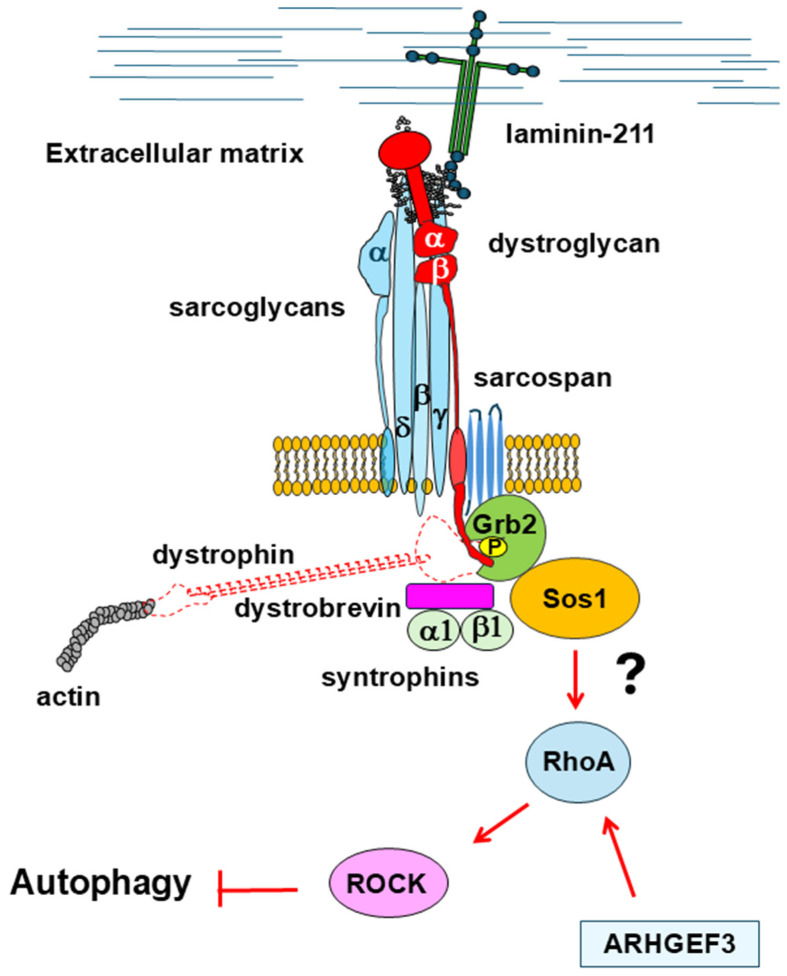
The binding of dystrophin and Grb2 to the C-terminus of β-dystroglycan is mutually exclusive. In the absence of dystrophin, this interaction is shifted toward Grb2 and β-dystroglycan binding. This can lead to the activation of the RhoA/ROCK signaling pathways that repress autophagy. The activation of RhoA can be further promoted by ARHGEF3, a RhoA guanine nucleotide exchange factor, whose expression level is enhanced in DMD.

**Figure 6 cells-14-00721-f006:**
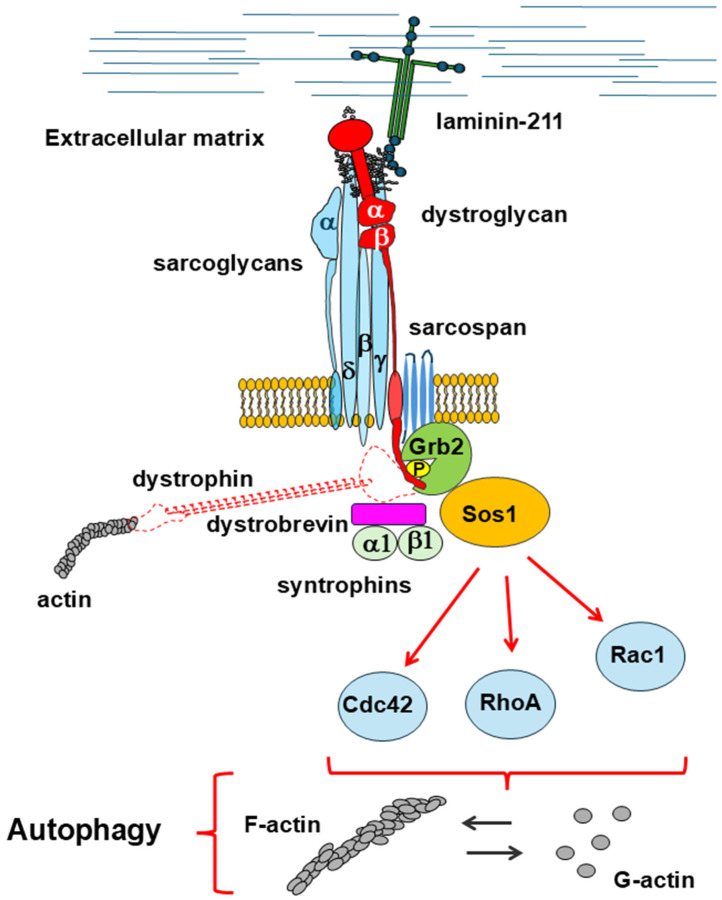
The interaction between β-dystroglycan and Grb2, which is stabilized in the absence of dystrophin, could lead to the activation of different GTPase proteins, such as RhoA, Rac1, and Cdc42, which interfere with actin polymerization. Different steps of autophagy could be affected by an altered equilibrium between G-actin and F-actin.

## Data Availability

No new data were created or analyzed in this study.

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
