# Peer review of "Misregulation of the Ubiquitin–Proteasome System and Autophagy in Muscular Dystrophies Associated with the Dystrophin–Glycoprotein Complex"

_cells, 2025, doi:10.3390/cells14100721_

Round 1
Reviewer 1 Report
Comments and Suggestions for Authors
In the present manuscript, Bozzi et al are providing a comprehesive review focusing in the description of the alteration of the ubiquitin and autophagy machineries for those conditions knows to as muscular dystrophies. Moreover they retraced in-depth reports describing how these dysfuctional processes could be downstream of the destabilization of the DAPC, by adding insightful hints for readers.
It is also well accepted that such alterions occurs in dystrophic mucles and, in some cases, it has been demonstrated that such autophagy alterations can be partially reverted/ameliorated by applying opportune dietary regimens or pharmacological agents mimicking caloric restriction. Including such information could be of values for reader.
Moreover, authors focused their attention in the description of reports that are well consolidated in the field while novel findings are not included. For instance some selective autophagy seem to be affected by DAPC destabilization or important for proper muscle behaviour. Recently ER-phagy has been demonstrated essential for proper sarcoplasmic reticulum maturation during differentiation. It could be of interest adding some pieces of literature to this direction, offering a new literature product and not a renewed copy of something existing
Author Response
Comments 1:
In the present manuscript, Bozzi et al are providing a comprehesive review focusing in the description of the alteration of the ubiquitin and autophagy machineries for those conditions knows to as muscular dystrophies. Moreover they retraced in-depth reports describing how these dysfuctional processes could be downstream of the destabilization of the DAPC, by adding insightful hints for readers.
It is also well accepted that such alterions occurs in dystrophic mucles and, in some cases, it has been demonstrated that such autophagy alterations can be partially reverted/ameliorated by applying opportune dietary regimens or pharmacological agents mimicking caloric restriction. Including such information could be of values for reader.
Response 1:
We wish to thank the Reviewer for their focused attention to this important point. Our original manuscript already reported on some studies highlighting the efficacy of a low-protein diet (ref. no. 63) and AMP-activated protein kinase (AMPK) activators in promoting autophagy in muscular dystrophy (refs. no. 64, 83-87). However, as pointed out by the Reviewer, we did not clearly state that AMPK agonists are pharmacological agents that mimic a condition of caloric restriction. To clarify, we have added the following sentence at lines 315-316 of the revised version: “mimic caloric restriction by stimulating the fuel-sensing AMP-activated protein kinase (AMPK)”, underlined in yellow.
Comment 2:
Moreover, authors focused their attention in the description of reports that are well consolidated in the field while novel findings are not included. For instance some selective autophagy seem to be affected by DAPC destabilization or important for proper muscle behaviour. Recently ER-phagy has been demonstrated essential for proper sarcoplasmic reticulum maturation during differentiation. It could be of interest adding some pieces of literature to this direction, offering a new literature product and not a renewed copy of something existing
Response 2:
We wish to thank the Reviewer for this suggestion. We have now included some recent articles that discuss impaired mitophagy in Duchenne muscular dystrophy, as well as the improvement of the dystrophic phenotype through pharmacological or genetic restoration of mitophagy (refs. no. 65-69 and 72). Additionally, we have quoted some findings (ref. no. 73) related to the role of ER-phagy in myoblast differentiation (see lines 272-287 of the revised version, underlined in yellow).
Reviewer 2 Report
Comments and Suggestions for Authors
The manuscript examines the different signaling pathways that modulate protein recycling, focussing mainly on muscular dystrophies and autophagy.
The review is clear, comprehesive and enlighting basic mechanisms of phenomena in degenerative processes in muscular dystrophies. the data are presented in an appropiate way.
The figures are nicely drawn and easy to interprete and understand. They enlighten the text.
The main statement of the manuscript is well defined and all the references cited underline the fact, that there is a misregulation of the ubiquitin-proteasome system and autophagy in muscular dystrophies assciated with the dystrophin-dystroglycan complex.
The illustrations are excellent and contribute to the clear understanding of current knowledge.
No changes needed.
Author Response
Comments:
The manuscript examines the different signaling pathways that modulate protein recycling, focussing mainly on muscular dystrophies and autophagy.
The review is clear, comprehesive and enlighting basic mechanisms of phenomena in degenerative processes in muscular dystrophies. the data are presented in an appropiate way.
The figures are nicely drawn and easy to interprete and understand. They enlighten the text.
The main statement of the manuscript is well defined and all the references cited underline the fact, that there is a misregulation of the ubiquitin-proteasome system and autophagy in muscular dystrophies assciated with the dystrophin-dystroglycan complex.
The illustrations are excellent and contribute to the clear understanding of current knowledge.
No changes needed.
Response:
We truly appreciate the comprehensive assessment of our work and wish to thank the Reviewer for their positive comments.